# RELATIVE PIXEL PREDICTION FOR AUTOREGRESSIVE IMAGE GENERATION

## ABSTRACT

In natural images, transitions between adjacent pixels tend to be smooth and gradual, a fact that has long been exploited in image compression models based on predictive coding. In contrast, existing neural autoregressive image generation models predict the absolute pixel intensities at each position, which is a more challenging problem. In this paper, we propose to predict pixels relatively, by predicting new pixels relative to previously generated pixels (or pixels from the conditioning context, when available). We show that this form of prediction fare favorably to its absolute counterpart when used independently, but their coordination under an unified probabilistic model yields optimal performance, as the model learns to predict sharp transitions using the absolute predictor, while generating smooth transitions using the relative predictor. Experiments on multiple benchmarks for unconditional image generation, image colorization, and super-resolution indicate that our presented mechanism leads to improvements in terms of likelihood compared to the absolute prediction counterparts.

## 1 INTRODUCTION

It is has long been appreciated in the field of image compression (Harrison, 1952; Haskell & Netravali, 1997) that adjacent pixels in natural images tend to share similar colors and intensities. This is evidenced in Figure 1, where we plot the marginal distribution of absolute sub-pixel (i.e., color channel) intensities and the difference between the sub-pixel intensities to the corresponding sub-pixel immediately to its left. While the absolute pixel values tend to be distributed non-trivially across the whole range of possible sub-pixel values $[0, 255]$ with peaks around the values of 100 and 255, the relative sub-pixel values closely resembles a Laplacian distribution with the mean at 0 as most pixels do not differ greatly from their neighbouring pixels (Takamura, 1996). This suggests that image generation process could mostly be done by predicting the relative distance between existing and new pixels, and absolute pixel prediction is only needed to generate more abrupt transitions, such as boundaries between two different objects. In predictive coding compression models, this fact is leveraged to achieve increased compression rates by learning to predict pixels as a delta between pixels rather than predicting their absolute values (al Mahmood & Al-Rubaye, 2014).

Autoregressive models are one of the main forces driving research in image generation van den Oord et al. (2016a); Salimans et al. (2017); Parmar et al. (2018); Chen et al. (2018); Child et al. (2019). In contrast to other popular image models based on adversarial methods (Goodfellow et al., 2014; Radford et al., 2015) or that use latent variables (van den Oord et al., 2017), autoregressive models have a tractable likelihood function that decomposes image generation into a sequence of conditionally dependent pixel predictions. This is a desirable property that provides the grounds for applying the same principles of predictive coding in the image generation process.

In this paper, we propose an image generation model where new pixels are generated by modeling the differences between new pixels and pre-existing ones. In contrast with existing approaches that tackle the challenging problem of learning to generate each pixel in terms of their absolute values (van den Oord et al., 2016b), we model gradually shifting pixel intensities in relative terms. From the modeling perspective, we propose a **copy and adjustment** mechanism, where new pixels are predicted by selecting an existing pixel, and adjusting its sub-pixel values to generate a new pixel. We show that new generation methodology generalizes better for multiple datasets in unconditional image generation and image translation tasks, namely colorization (Cheng et al., 2016; Zhang et al., 2016;

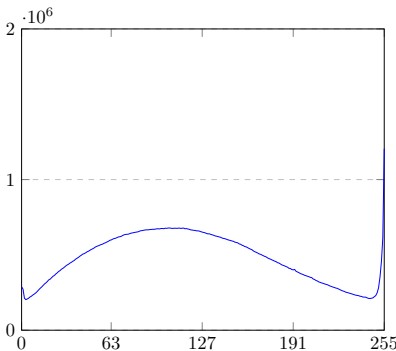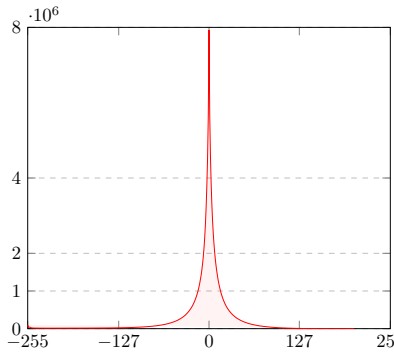

Figure 1: Marginal distribution of absolute and relative (to the left pixel) pixel intensities in the training set of CIFAR-10.

Guadarrama et al., 2017; Iizuka et al., 2016; Guadarrama et al., 2017) and super-resolution (Parmar et al., 2018). Additionally, we show that our mechanism can be adapted to copy and adjust pixels from the input image in image-to-image translation tasks. Finally, we show that the different generation mechanisms can be unified into a single loss function with tractable likelihood computation.

The paper is organized as follows: Firstly, we describe our copy and adjustment model (§2). Then, we describe its application in image-to-image translation tasks (§3). Afterwards, we describe a mixture model that combines different forms of pixel prediction (§4). Then, we provide experimental results that support our claims (§5). Finally, we describe the positioning of our work with respect the previous research (§6) and proceed to the conclusions (§7).

## 2 RELATIVE PIXEL PREDICTION

PixelCNN (van den Oord et al., 2016a) models the generation of an image $\boldsymbol{x} = (\mathbf{x}_1, \mathbf{x}_2, \ldots, \mathbf{x}_{n^2})$ with width and height $n$ as a probability density function $p(\boldsymbol{x})$ that is factorized into an autoregressive prediction of pixels as follows[1]:

$$p(\boldsymbol{x}) = \prod_{i=1}^{n^2} p(\mathbf{x}_i \mid \boldsymbol{x}_{k<i}), \tag{1}$$

where $\mathbf{x}_i$, which consists of $C$ subpixel channels $(x_{i,1}, x_{i,2}, \ldots, x_{i,C})$, is conditioned on $k$ previously generated pixels. Each pixel prediction $p(\boldsymbol{x}_i \mid \boldsymbol{x}_{k<i})$ is further factorized by the prediction of its $C$ subpixel channels,

$$p(\mathbf{x}_i \mid \boldsymbol{x}_{k<i}) = \prod_{c=1}^{C} p(x_{i,c} \mid \mathbf{x}_{i,m<c}, \boldsymbol{x}_{k<i}). \tag{2}$$

Similarly, each new sub-pixel is conditioned on the previously $k$ generated pixels and the $m$ generated sub-pixels. The probability density function of each sub-pixel $x_{i,c}$ is modelled by bucketing the range of continuous values in the range $[0, 1]$ into 256 evenly distributed classes (van den Oord et al., 2016a), which is modeled as a categorical distribution over $V_a = [0, 255]$. More formally, given a hidden state $\mathbf{h}_{i,c-1}$, obtained by composing the history of generated sub-pixels, the following pixel is generated as follows:

$$p(x_{i,c} \mid \mathbf{x}_{i,m<c}, \boldsymbol{x}_{k<i}) = \text{softmax}_{V_a}(\mathbf{W}_s \mathbf{h}_{i,c-1}). \tag{3}$$

Our work changes the probability density function in Equation 2, which computes the subpixel values directly, with one that computes these values relative to previously generated pixels. More formally, rather than predicting $x_{i,c}$, we compute the distance $\Delta_{x_{r,c}}^{x_{i,c}} = x_{i,c} - x_{r,c}$, where $\mathbf{x}_r$ denotes the previously generated pixel at position $r$. Under this formulation, we rewrite Equation 2 as follows:

---

[1]Sequences of pixels representing either an image of part of an image are written as $\boldsymbol{x}$, a vector representing a multi-channel pixel is written $\mathbf{x}$, and scalars are written $x$. The $i$th scalar element of vector $\mathbf{x}$ is $x_i$, and the $j$th vector element of a sequence $\boldsymbol{x}$ is $\mathbf{x}_j$.

$$p(\mathbf{x}_i \mid \boldsymbol{x}_{k<i}) = p(\mathbf{x}_r, \Delta_{x_{r,c}}^{x_{i,*}} \mid \boldsymbol{x}_{k<i}) = p(\mathbf{x}_r \mid \boldsymbol{x}_{k<i}) \prod_{c=1}^{C} p(\Delta_{x_{r,c}}^{x_{i,c}} \mid \mathbf{x}_r, \boldsymbol{x}_{i,m<c}, \boldsymbol{x}_{k<i}). \quad (4)$$

Illustrated in Figure 2, our relative pixel prediction can be interpreted as selecting a pixel to copy under the probability mass function $p(\mathbf{x}_r \mid \boldsymbol{x}_{k<i})$ and then predicting the adjustment needed to add to that pixel to obtain $\mathbf{x}_i$ modelled by $p(\Delta_{x_{r,c}}^{x_{i,c}} \mid \mathbf{x}_r, \mathbf{x}_{i,m<c}, \boldsymbol{x}_{k<i})$. We name the former the copy mechanism and the latter the adjustment mechanism.

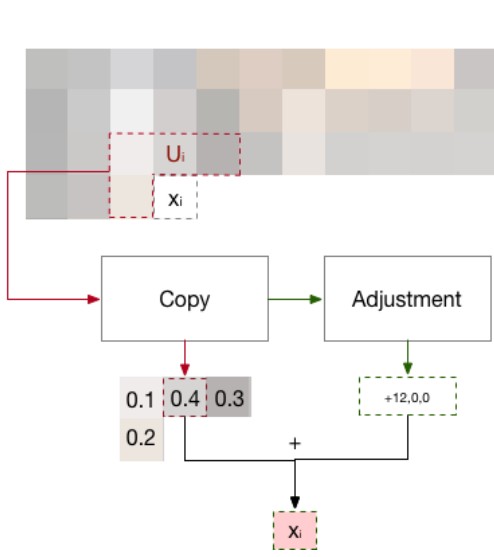

Figure 2: Illustration of the copy and adjustment mechanism. The copy mechanism selects a previously generated pixel and the adjustment mechanism modifies the pixel to generate the next pixel.

## 2.1 COPY MECHANISM

The concept of learning to reference a pre-existing pixel under a given distribution $p(\mathbf{x}_r \mid \mathbf{x}_{k_i})$ is closely related to the copy mechanism frequently used in natural language generation models (Ling et al., 2016; Gu et al., 2016; Merity et al., 2016), where observed words, either from the input or previously generated output, can be copied to generate new outputs using a pointer network (Vinyals et al., 2015). Pointer networks define a probability distribution over a discrete space of units, in our case previously generated pixels, and compute $p(\mathbf{x}_r \mid \boldsymbol{x}_{k<i})$ as follows:

$$u_{r'} = \mathbf{v}^T \tanh(\mathbf{W}_s^a \mathbf{h}_{i,0} + \mathbf{W}_r^a \mathbf{h}_{r'+1,0} + \mathbf{W}_x^a \mathbf{x}_{r'}) \quad (5)$$

$$p(\mathbf{x}_r \mid \boldsymbol{x}_{k<i}) = \text{softmax}_{U_i}(\mathbf{u}), \quad (6)$$

where $u_{r'}$ (for each $r'$ in $U_i$) computes the affinity between current state $\mathbf{h}_{i,0}$ before any sub-pixels at position $i$ are generated and state of a previously generated pixel $\mathbf{h}_{r'+1,0}$ after all pixels in candidate source position $r'$ have been generated with sub-pixel values $\mathbf{x}_r$. Parameters $\mathbf{W}_s^a, \mathbf{W}_r^a, \mathbf{W}_x^a$ and $\mathbf{v}$ are used to model this function.

The probability of choosing a given pixel $\mathbf{x}_r$ is computed as a softmax over the affinities of all pixels that can be selected $U_i$. While applications of pointer networks in natural language processing can be applied to whole sentences, which contain perhaps 40–200 words, the large number of pixels in images (e.g. 4096 in 64×64 images) makes this practice prohibitive. Following (Parmar et al., 2018), we restrict this choice to a predefined set of neighbouring pixels.

Finally, near the edges of images, some positions in $U_i$ will lie outside of the image. To address this case, we always can copy the pixel with value $\mathbf{x}_{\text{null}} = [0, 0, 0]$ and learn a special state $\mathbf{h}_{\text{null}}$. This is needed for the top and left edges of the image, where no pixels have been previously generated.

## 2.2 Adjustment Mechanism

The adjustment mechanism aims to learn a continuous variable $\Delta_a^b$ which measures the absolute distance between two sub-pixels $a$ and $b$. In order to keep log-likelihood computations consistent with the buckets defined in previous work (van den Oord et al., 2016a), we compute the distances over the discretized pixel values. Thus, we model the distance function as a categorical distribution over integers in the range $V_r = [-255, 255]$. This quantifies the number of buckets that need to be adjusted with relation to the pixel to be copied. The computation of the distance distribution is performed as follows:

$$\mathbf{a} = \mathbf{W}_s^p \mathbf{h}_{i,c-1} + \mathbf{W}_r^p \mathbf{h}_{r,C} + \mathbf{W}_x^p \mathbf{x}_r \tag{7}$$

$$p(\Delta_{x_{r,c}}^{x_{i,c}} \mid \mathbf{x}_r, \mathbf{x}_{i,m<c}, \boldsymbol{x}_{k<i}) = \text{softmax}_{V_r}(\mathbf{a}). \tag{8}$$

Thus, the softmax that computes the required adjustment depends on the current hidden state $\mathbf{h}_{i,c-1}$, and the state and the pixel to copy from $\mathbf{h}_{r,C}$ and its values $\mathbf{x}_r$. It is worth mentioning that this softmax can be replaced by any mechanism that defines a categorical distribution, such as that described in (Salimans et al., 2017).

While the outcome of the relative predictor consists of a value in the interval $[-255, 255]$, for a given copied pixel, the number of valid adjustment values that situate the intensity within the valid range $[0, 255]$ remains as 256. For instance, if the copied pixel's value is 100, the range of possible adjustment values is $[-100, 155]$. While the model can learn to avoid such mistakes from the data, any probability mass assigned to invalid adjustment values is penalized in terms of perplexity. Thereby, we ensure that the model avoids such errors by limiting softmax defined in Equation 7 to the range of valid values.

## 2.3 Marginalization over Relative Pixels

For likelihood computation, the choice of the relative pixel $x_r$ is unobserved and must be inferred during training. As we define a small range of possible pixels for $x_r$, limited to neighbouring pixels, the marginal likelihood over the latent domain $U_i$ is tractable and can be defined as follows:

$$p(\mathbf{x}_i \mid \boldsymbol{x}_{k<i}) = \sum_{\mathbf{x}_r \in U_i} p(\mathbf{x}_r, \Delta_{x_{r,c}}^{x_{i,1}}, \ldots, \Delta_{x_{r,c}}^{x_{i,C}} \mid \boldsymbol{x}_{k<i})$$

$$\sum_{\mathbf{x}_r \in U_i} p(\mathbf{x}_r \mid \boldsymbol{x}_{k<i}) \prod_{c=1}^{C} p(\Delta_{x_{r,c}}^{x_{i,c}} \mid \mathbf{x}_r, \mathbf{x}_{i,m<c}, \boldsymbol{x}_{k<i})$$

For generation, we first sample the pixel to copy $\mathbf{x}_r$ from the distribution $p(\mathbf{x}_r \mid \boldsymbol{x}_{k<i})$, then we sample an adjustment from the distributions $p(\Delta_{x_{r,c}}^{x_{i,c}} \mid \mathbf{x}_r, \mathbf{x}_{i,m<c}, \boldsymbol{x}_{k<i})$ for each sub-pixel value $c$, thereby generating $\mathbf{x}_i$.

## 3 Image-To-Image Translation

If the image generation process is conditioned on an input image $\boldsymbol{y}$, the copy mechanism can also be applied to copy pixels from the input image. This is desirable in tasks, such as image colorization and super-resolution, where the input image is structurally similar to the output image and we hypothesize that learning the adjustment in such tasks generalizes better than learning to generate from scratch.

PixelCNN can be adjusted for image translation by conditioning on a fully visible input image $\boldsymbol{y}$. More formally, each pixel $p(\mathbf{x}_i \mid \boldsymbol{x}_{k<i}, \boldsymbol{y})$ is now generated as follows:

$$p(\mathbf{x}_i \mid \boldsymbol{x}_{k<i}, \boldsymbol{y}) = \text{softmax}_{V_a}(\mathbf{W}_s \mathbf{h}_{i,c-1} + \mathbf{W}_m \mathbf{g}_i)$$

where $\mathbf{g}_i$ is the hidden state of the input image $y$ at pixel $i$. This differs from the output image hidden states $\mathbf{h}_{i,m}$ in that $\mathbf{g}_i$ has visibility of the whole input image.

Relative pixel prediction over the input image can be embedded by reformulating Equations 5 and 7 as follows:

$$u_{r'} = \mathbf{v}^T \tanh(\mathbf{W}_s^a \mathbf{h}_{i,0} + \mathbf{W}_m^a \mathbf{g}_{r'} + \mathbf{W}_y^a \mathbf{y}_{r'}) \tag{9}$$

$$p(\mathbf{x}_r \mid \boldsymbol{x}_{k<i}, \boldsymbol{y}) = \mathrm{softmax}_{B_i}(\mathbf{u}) \tag{10}$$

$$\mathbf{a} = \mathbf{W}_s^p \mathbf{h}_{i,c-1} + \mathbf{W}_r^p \mathbf{g}_r + \mathbf{W}_y^p \mathbf{y}_r \tag{11}$$

$$p(\Delta_{y_{r,c}}^{x_{i,c}} \mid \mathbf{x}_r, \mathbf{x}_{i,m<c}, \boldsymbol{x}_{k<i}, \boldsymbol{y}) = \mathrm{softmax}_{V_r}(\mathbf{a}), \tag{12}$$

where we replace the output pixels and their hidden states of the output pixels with those of the input, namely $\boldsymbol{y}$ and $\mathbf{g}$. Secondly, we define $B_i$ as the set of pixels in the input image $y$ that are available to be copied from position $i$. For simplicity, we also assume that $\boldsymbol{y}$ and $\boldsymbol{x}$ have the same dimensions. For image colorization, where the input image has only one channel, we simply copy the same channel three times. For super-resolution, we ensure this by first increasing to the desired dimensions using a heuristic method (e.g. bilinear interpolation). These do not correspond to optimal architecture choices for these tasks, but still allow the fair evaluation between the absolute and relative predictors.

## 4 UNIFIED PREDICTOR

The copy and adjust mechanism is self-sufficient for generating any image, and benefits from a denser distribution when neighbouring pixels share similar sub-pixel values. However, when large color perturbations are observed and no correlations between neighbouring pixels can be found, using this mechanism is intuitively disadvantageous, since the range of differences between two sub-pixel values is larger than the range of sub-pixel values (511 vs. 256). Furthermore, in image-to-image tasks it would also be desirable to combine both forms of relative prediction (e.g. copying from the input and output). Our solution to supporting both absolute and relative generation mechanisms is using latent predictors (Ling et al., 2016). We define our set of predictors as $P = [P_{abs}, P_{i\text{-}rel}, P_{o\text{-}rel}]$, where $P_{abs}$ denotes the absolute pixel predictor computed according to Equation 2 and $P_{i\text{-}rel}$ and $P_{o\text{-}rel}$ denote the relative pixel predictors defined in Equation 4 and 11. At each generation step $i$, we define the predictor probability $p(\pi \mid \boldsymbol{x}_{k<i})$ as follows:

$$p(\pi \mid \boldsymbol{x}_{k<i}) = \mathrm{softmax}_{[P_{abs}, P_{rel\text{-}i}, P_{rel\text{-}o}]}(\mathbf{W}_l \mathbf{h}_i), \tag{13}$$

where $\mathbf{W}_l$ defines the parameters used by the gating system that switches between the absolute and relative predictors.

It is also beneficial to prune marginally less frequent adjustment values in the relative predictor by removing them from the softmax in Equations 7, as these generally correspond to sharp transitions between pixels, which are better handled by the absolute predictor. We do this by pruning the domain of the softmax over the interval $[-255, 255]$ to the set of the $k$ most frequently observed adjustment values in the training data, where $k$ is a hyperparameter tuned on the validation set. This means that for pruned adjustment values, the relative predictor will predict the pixel with probability 0, which is acceptable in the unified predictor, as the absolute predictor would present a valid probabilistic interpretation for such pixels.

## 5 EXPERIMENTS

### 5.1 SETUP

Our experiments were performed on existing benchmarks for image generation, namely, CIFAR-10 (Krizhevsky, 2009) and downsampled ImageNet (Deng et al., 2009; Chrabaszcz et al., 2017).

For unconditional image generation, hidden states $\mathbf{h}$ are obtained using gated convolutional layers (van den Oord et al., 2016a). More concretely, after discrediting the input image $\boldsymbol{x}$, we apply a convolutional layer with kernel size 7 and hidden size $H_1$. Then, we apply an additional $D$ layers with kernel size 5 and hidden size $H_1$, with the exception of the final layer which has the hidden size

Table 1: Quantitative results using relative pixel prediction. Each cell reports the negative log-likelihoods in bits/dim on CIFAR-10's test set and ImageNet's validation set for unconditional image generation (U), colorization (C) and super-resolution (S). Row 0 is the baseline. Rows 1 and 2 describe the results obtained using only the input and output predictors, respectively. Rows 3 and 4 combines them with the absolute predictor. Row 5 denotes the combination of the input and output predictors.

| | | CIFAR-10 32×32 | | | IMAGENET 32×32 | | | IMAGENET 64×64 | | |
|---|---|---|---|---|---|---|---|---|---|---|
| | | U | C | S | U | C | S | U | C | S |
| 0 | ABSOLUTE | 3.02 | 1.25 | 2.87 | 3.83 | 2.17 | 3.74 | 3.55 | 2.02 | 3.49 |
| 1 | RELATIVE(O) | **2.99** | 0.99 | 2.82 | 3.80 | 2.08 | 3.71 | 3.53 | **1.63** | 3.41 |
| 2 | RELATIVE(I) | - | 1.21 | 2.85 | - | 2.16 | 3.73 | - | 1.99 | 3.46 |
| 3 | ABSOLUTE + RELATIVE(O) | 3.00 | 1.03 | 2.83 | **3.79** | 2.11 | **3.69** | 3.52 | 1.69 | 3.40 |
| 4 | ABSOLUTE + RELATIVE(I) | - | 1.20 | 2.84 | - | 2.14 | 3.72 | - | 1.98 | 3.46 |
| 5 | RELATIVE(I+O) | - | **0.97** | 2.81 | - | **2.06** | 3.70 | - | **1.63** | **3.38** |

of $H_2$. Furthermore, we add a skip connection (Ronneberger et al., 2015), between each intermediate layer to the final layer. This final layer corresponds to the hidden states **h**. For CIFAR-10, we set $D = 15$, $H_1 = 64$ and $H_2 = 512$, and train with a batch size of 32. For ImageNet, we set $D = 20$, $H_1 = 256$ and $H_2 = 2048$ and train with a batch size of 128.

For image-to-image translation, we apply the operations to both $x$ and $y$ to obtain **h** and **g**, respectively. However, as $y$ is fully observed, no mask is applied in the convolutions to obtain **g**. For CIFAR-10, we set $D = 15$, $H_1 = 128$ and $H_2 = 512$, and train with a batch size of 32. For ImageNet, we set $D = 20$, $H_1 = 256$ and $H_2 = 2048$ and train with a batch size of 64. For colorization, we remove the colorization of the original image. For super-resolution, we follow the approach applied in (Dahl et al., 2017), and reduce resolution of the image to 8×8. Evaluation is performed in terms of negative log-likelihood measured in bits/dim[2].

As for the prediction mechanisms, we compare the baseline that relies solely on absolute predictors against the best setups found on CIFAR-10 from our ablation study (Appendix A), which is carried out on the validation set. For unconditional image generation, we use a relative predictor that can copy from the pixel to the left and top of the generated pixel. We report results, when only relative prediction is used and when both absolute and relative predictors are used. Here, the pruning threshold is set to 20%, which is also chosen based on the results of the ablation test. For image-to-image translation, the same setup is used when only relative predictors that copy from the output is used. When we consider the input image as another source that can be copied from, we define a $3 \times 3$ grid around the position of the output pixel, and any corresponding pixel in the input image that falls within that grid can be selected as a target to be copied from.

## 5.2 QUANTITATIVE RESULTS

Table 1 reports the results on the test set of CIFAR-10 and validation sets of ImageNet. Our baseline implementation of PixelCNN (row 0) achieves a negative log-likelihood of 3.02 bits/dim. From rows 1 and 2, we observe that both relative predictors yield improvements over the baseline. Overall, combining each relative form of prediction with the absolute predictor (rows 3 and 4) tends to work well for all tasks except for colorization, where combining the output relative predictor and absolute predictors tend to yield lower results. Finally, combining the input and output relative predictors yields the optimal overall results (row 5). We do not illustrate results when all predictors are combined, as the presence of the input predictor and the absolute predictor causes the latent predictor network (Ling et al., 2016) to ignore the absolute predictor, and thereby, results are identical to that in row 5.

---

[2]We acknowledge that extensive work has been conducted in the tasks of super-resolution (Ledig et al., 2016) and colorization (Nazeri & Ng, 2018), which are evaluated on different metrics, we refrain from directly comparing our model to the aforementioned work as our main motivation is to measure the effectiveness of the relative pixel prediction mechanism with respect to copying pixels from the input.

Figure 3: 32×32 images generated by our model on CIFAR-10. Columns correspond to the input of the model, the sample obtained from the model, the sampled latent variables and the reference. Three colors are used for the latent variables: White → absolute predictor; Blue → relative predictor from the left pixel and Green → relative predictor from the top pixel.

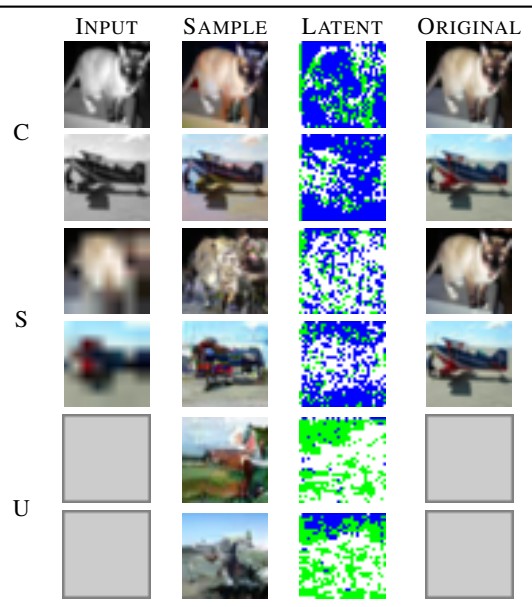

## 5.3 LATENT VARIABLE ANALYSIS

For a more in-depth understanding of the relative and absolute prediction choices, we provide samples and latent variable choices for CIFAR-10 validation set in Table 3. Columns "Input", "Sample" and "Reference" denote the input observed by the model, the sample that it generates, and the expected result, respectively. In the column denoted as "latent", we illustrate the latent choices made by the model during the sampling process, where white pixels correspond to the absolute predictor, and blue and green pixels correspond to the relative predictor that copies from the left and top pixels, respectively.

We observe that the majority of the images are generated using the relative prediction mechanism from the left pixel, with top pixels only used when the left pixel is unavailable (e.g. left edge of the image) or vertically oriented objects (e.g. legs of the cat). However, there are some instances where images are predominantly predicted by copying from the top pixel. In simpler tasks, such as colorization, the model seems to rarely use the absolute predictors applying it only to generate the transitions between objects. In more advanced tasks, such as super-resolution and unconditional image generation, relative predictors are often applied to generate continuous textures (e.g. sky and glass) whereas more complex objects such as houses and trees are generated by combining both absolute and relative predictors.

## 5.4 IMPACT ON STATE-OF-THE-ART MODELS

While we show that relative pixel prediction can improve the quality of the predictions of the model presented in (van den Oord et al., 2016a), state-of-the-art models can achieve significantly better results by incorporating multiple modeling innovations (Salimans et al., 2017; Parmar et al., 2018; Chen et al., 2018). Thus, to investigate if the work proposed can be used to further augment state-of-the-art models. We build upon the model described in (Razavi et al., 2019), which defines a 30-layer neural network with transformer layers. We replace the discretized logistic mixture of components with 32 components with a relative pixel prediction that copies from the left and top of the desired pixel with, whose output is pruned with a 20% threshold. Finally, we also use outcome masking. Results are illustrated in Table 2, where we can see that our model fares favourably with state-of-the-

Table 2: Comparison between state-of-the-art models in unconditional image generation on CIFAR-10. Cells report log-likelihood scores measured in bits/dim.

|  | CIFAR-10 |
| --- | --- |
| DRAW (GREGOR ET AL., 2015) | 4.10 |
| GROW (KINGMA & DHARIWAL, 2018) | 3.35 |
| MINTNET (SONG ET AL., 2019) | 3.32 |
| PIXELCNN (VAN DEN OORD ET AL., 2016A) | 3.03 |
| PIXELCNN++ (SALIMANS ET AL., 2017) | 2.92 |
| TRANSFORMER (PARMAR ET AL., 2018) | 2.90 |
| PIXEL-SNAIL (CHEN ET AL., 2018) | 2.85 |
| DELTA VAE (RAZAVI ET AL., 2019) | 2.83 |
| RELATIVE PIXELCNN (THIS WORK) | 2.82 |
| SPARSE TRANSFORMER (CHILD ET AL., 2019) | 2.80 |

art models in this task. Improvements are more marginal (from 2.83 to 2.82) but supplements the upward trend from experiments using the vanilla PixelCNN.

## 6 RELATED WORK

Copy mechanisms have been widely used in sequence-to-sequence models (Sutskever et al., 2014) in order to copy tokens from the input sequence (Ling et al., 2016; Gu et al., 2016), previously generated tokens (Merity et al., 2016), or from structured data, such as databases (Yang et al., 2016; Ahn et al., 2016). The basic copy mechanism recalls pointer networks (Vinyals et al., 2015) but extends it by allowing the copied element to be modified. A similar approach can be found in (Ling et al., 2017), where copied tokens can be modified with multiple operations before generation. Image compression techniques encode deltas between pixels rather than absolute intensities (Harrison, 1952), and even natural visual systems code deltas rather than absolute magnitudes Rao & Ballard (1999).

A major force driving progress on auto-regressive image generation models is the advancements of self-attention mechanisms (Parmar et al., 2018; Child et al., 2019), which allows better modeling of long range dependencies within images. However, improving generalization by incorporating better prediction mechinism has also been explored in the past. The usage of discretized logistic mixture likelihoods (Salimans et al., 2017), where the independence between consecutive pixel intensities is alleviated here, which leads to be better generalization is an example of such exploits. Our work is another instance of such lines of research, where we incorporate a mechanism that exploits the fact that consecutive pixels tend to share similar values.

Another major driver of progress in image generation is the trend of adversarial image generation (Goodfellow et al., 2014; Isola et al., 2016; Karras et al., 2017; Brock et al., 2018; Rivière et al., 2019; Wang & Huan, 2019). While our model is not directly applicable to this set of models, as our proposed work operates on explicit probability estimates of pixel distributions rather than implicit ones, the concept of relative pixel prediction is likely to be beneficial to these models.

## 7 CONCLUSION

We presented an alternative parameterization for autoregressive models for image generation, where pixels are generated relative to pre-existing pixels. This is achieved by learning two probability mass functions over what pixel to copy and how the copied pixel must be adjusted to obtain the new pixel. We show that generating new pixels as deltas from previously generated neighbouring pixels has better generalization properties compared to existing methods that generate pixels in absolute terms. Finally, we propose a mechanism that combines the multiple types of predictions into mixture model, which allows exact inference at training and decoding time. Experimental results show that our proposed models achieve improvements to their absolute counterparts, and can achieve competitive results compared to state-of-the-art models in unconditional image generation.

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

Table 3: Ablation study over different predictors. Each cell reports the negative log-likelihoods in bits/dim on the CIFAR-10 for unconditional image generation (U), colorization (C) and super-resolution (S). Row 0 is the baseline. Rows 1–3 denote results using only relative predictors from the left pixel, top pixel and both, respectively. Rows 3 and 4 allow the usage of a 2×2 and 3×3 grid centered around the predicted pixel. Rows 6–8 combine the absolute predictor with the relative one, and rows 9–11 use a pruned distance list for the relative predictors. Rows 12–14 use only the relative predictor that copies from the input pixel in the same position as the predicted pixel, and in a 2×2 and 3×3 grids around that position. Rows 15–17 denote the combinations between the absolute and relative predictor from the input. Finally, we combine all the aforementioned predictors in rows 18–19.

| | | U | C | S |
|---|---|---|---|---|
| 0 | ABSOLUTE | 3.02 | 1.23 | 2.86 |
| | RELATIVE(O) | | | |
| 1 | LEFT PIXEL | 3.00 | **0.96** | 2.84 |
| 2 | TOP PIXEL | 3.00 | 0.98 | 2.83 |
| 3 | LEFT+TOP PIXEL | **2.99** | 1.00 | **2.80** |
| 4 | 2×2 | 3.00 | 1.06 | 2.82 |
| 5 | 3×3 | 3.01 | 1.05 | 2.83 |
| 6 | (3) + (0) | 3.01 | 1.22 | 2.85 |
| 7 | (4) + (0) | 3.02 | 1.26 | 2.86 |
| 8 | (5) + (0) | 3.02 | 1.29 | 2.86 |
| 9 | (3P) + (0) | 3.00 | 1.01 | **2.80** |
| 10 | (4P) + (0) | 3.00 | 1.00 | 2.81 |
| 11 | (5P) + (0) | 3.00 | 1.00 | 2.81 |
| | RELATIVE(I) | | | |
| 12 | CENTER PIXEL | - | 1.20 | 2.91 |
| 13 | 2×2 | - | 1.19 | 2.85 |
| 14 | 3×3 | - | 1.19 | **2.84** |
| 15 | (12) + (0) | - | 1.21 | 2.86 |
| 16 | (13) + (0) | - | 1.18 | 2.86 |
| 17 | (14) + (0) | - | **1.17** | 2.85 |
| | RELATIVE(I+O) | | | |
| 18 | (3P) + (14) | - | **0.94** | **2.78** |
| 19 | (3P) + (14) + (0) | - | 0.98 | 2.79 |
| | MASKED RELATIVE(O) | | | |
| 20 | (3PM) | **2.98** | 0.95 | 2.79 |
| | MASKED RELATIVE(I+O) | | | |
| 21 | (3PM) + (14) | - | **0.93** | **2.78** |
| 22 | (3PM) + (14) + (0) | - | 0.96 | 2.79 |

## A    ABLATION STUDY

To test the effect of different components described in our work, we start by performing an ablation test over different setups on CIFAR-10, and select the setups that are most promising to be applied to the larger datasets. This study is reported on Table 3. Columns U, C and S report the negative log-likelihood scores in bits/dim for unconditional image generation, colorization and super-resolution, respectively. Each numbered row uses a different set of predictors starting from our baseline implementation of PixelCNN (van den Oord et al., 2016a) in row 0.

**Output Relative Prediction**    Rows 1 and 2 show the results obtained by computing the adjustment score in Equation 7 with respect to the left or top pixels, respectively. As the choice of which pixel to select as the reference for the adjustment is predetermined, the copy mechanism is not required. Surprisingly, this simple change yields improvements in multiple datasets compared to absolute predictors. On average, adjusting from the left pixel yields more improvement compared to adjusting the top pixel, yet differences seem to be minor. Combining both predictors tends to improve the scores of the model, as observed in row 3, with the exception of the colorization task. In row 4, we also observe that adding all pixels within an 2×2 grid apart from the generated pixel (top, left and diagonals) does not yield additional improvements. The same occurs if we extend this grid to two

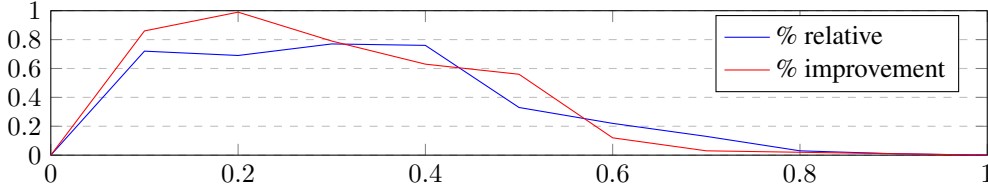

Figure 4: Effects of the pruning threshold when both absolute and relative predictors are used. The X-axis denotes the percentage of values that kept in the relative predictor's softmax.

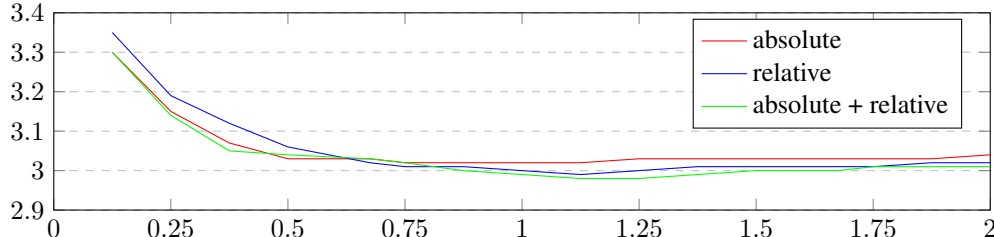

Figure 5: Effects of changing the number of parameters in the model using the absolute (red line) and relative predictors (blue line). The $x$-axis denotes the fraction of the parameters used in the hidden layers, and the $y$-axis reports the log-likelihood obtained on the CIFAR-10 validation set for unconditional image generation.

positions, as shown in row 5. We believe this is because neighbouring pixels tend to share similar colors, so more additional pixels are unlikely to add any additional information than the left and top pixels.

Rows 6, 7 and 8 illustrate the results obtained by combining the absolute predictor from row 0 with each of the relative predictors in rows 3, 4, and 5. Such setups cause the latent predictor networks to assign most probability mass to the absolute predictor. This suggests that the model is over-fitting to the absolute predictor, which has a faster convergence rate due to its smaller softmax (256 vs. 511).

**Thresholding Relative Prediction** To effectively combine the absolute and relative predictors we pruned the softmax of the adjustment mechanism to the $k$ most observed values in the training set (as discussed in §4). Rows 9, 10 and 11 report the results obtained when we set the pruning threshold to 20%, keeping only the 20% most frequent adjustment values in the training set. In CIFAR-10, we obtain results that are close to the best performing setups. In Figure 4, we study the results obtained using different threshold values in CIFAR-10. The red line represents the relative improvement obtained for unconditional image generation in the validation set using the model setup corresponding to row 9, and we can observe the highest improvement at 20%. We also report the percentage of pixels generated using the relative prediction mechanism when generating 100 samples with the blue line. We observe that while at lower thresholds the relative prediction mechanism is used approximately for the generation of 70% of images, when this value approaches 100%, this relative prediction is rarely used, explaining the lack of improvements without pruning.

**Parameter Efficiency** Figure 5 illustrates the log-likelihood scores for unconditioned image generation for different hyper-parameters for the hidden layer sizes, namely $H_1$ and $H_2$. We can observe that both absolute (red line) and relative (blue line) loss functions start under-performing as we increase the size of the model. Yet, the relative improvement of our loss function is maintained. However, the relative predictor requires more parameters to start performing effectively, if we drop the number of parameters to half, the absolute predictor exhibits a superior performance. We believe that this is because the relative predictor requires more parameters to learn the harsher transitions between pixels. Finally, if we combine the absolute and pruned relative predictors, the model achieves lower overall log-likelihood scores across different parametrizations.

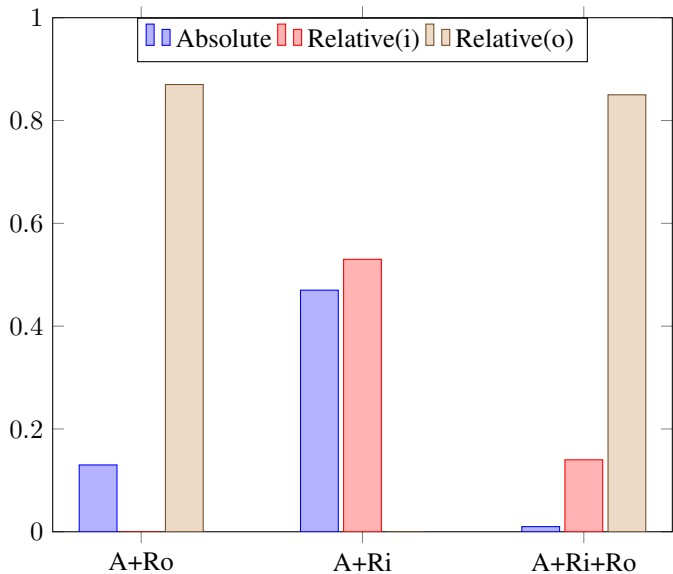

Figure 6: Predictor usage in different setups for colorization in CIFAR-10. Each bar denotes the percentage of usages after sampling once for each image in the validation set.

**Input Relative Prediction** For relative pixel prediction from the input, we tested models that copy from the center pixel, then in a 2×2 grid and 3×3 grid centered at the position of the pixel to be predicted. Results are reported in rows 12, 13 and 14. Here, we observe that a large patch of pixels to copy from benefits the model. Interestingly, we did not encounter any problems combining the absolute and relative predictors when copying from the input in these tasks. We believe this is because the transitions between the input and output pixels in these tasks is more straight-forward, whereas the transition between neighbouring pixels can contain harsh transitions (e.g. edges between objects), which take longer to learn. Results that combine the input relative predictor and absolute predictor are shown in rows 15, 16 and 17. Here, we observe that combining predictors yields in marginally better scores on average.

**Predictor Combination** Rows 18 and 19 show the improvements obtained by combining both input and output predictors, where we observe that optimal results exclude the usage of an absolute predictor. This suggests that the information that the two forms of relative prediction are complementary for the tasks that are tested. That is, the output relative prediction learns the softer transitions between continuous surfaces, whereas input relative prediction learns the harsher transitions between objects. Figure 6 illustrates the percentage to usages of the absolute predictor, input relative predictor and output relative predictor for the colorization task. We observe that the absolute predictor is applied considerably when used individually with either of the relative predictors (A+Ro and A+Ri), while rarely used with both predictors (A+Ri+Ro).

**Outcome Masking** Rows 20, 21 and 22 show the improvements that can be obtained by masking invalid entries in the relative predictors. While improvements are marginal, we can observe that perplexities are consistently better using masking, as we are removing invalid outcomes that are certainly leading to losses in probability mass.

## B  SAMPLED IMAGES

Examples of sampled images are illustrated in Figure 7. Columns "Absolute" "Left Pixel Only" and "Absolute + Relative" refer to the models trained using setups 0, 1 and 9 in Table 3. Here, we can observe that the proposed models can capture the richness of the image distribution from the data. Interestingly, sharp transitions are also captured when all pixels are predicted relatively from the pixel to the left (Column "Left Pixel").

Figure 7: 32×32 images generated by models trained on CIFAR-10 with using the absolute predictor (Column "Absolute"), a relative predictor that always copies from the left pixel (Column "Left Pixel Only") and the combination of the absolute predictor and a relative predictor that copies from the top and left pixels (Column "Absolute+Relative").

ABSOLUTE      LEFT PIXEL ONLY      ABSOLUTE + RELATIVE

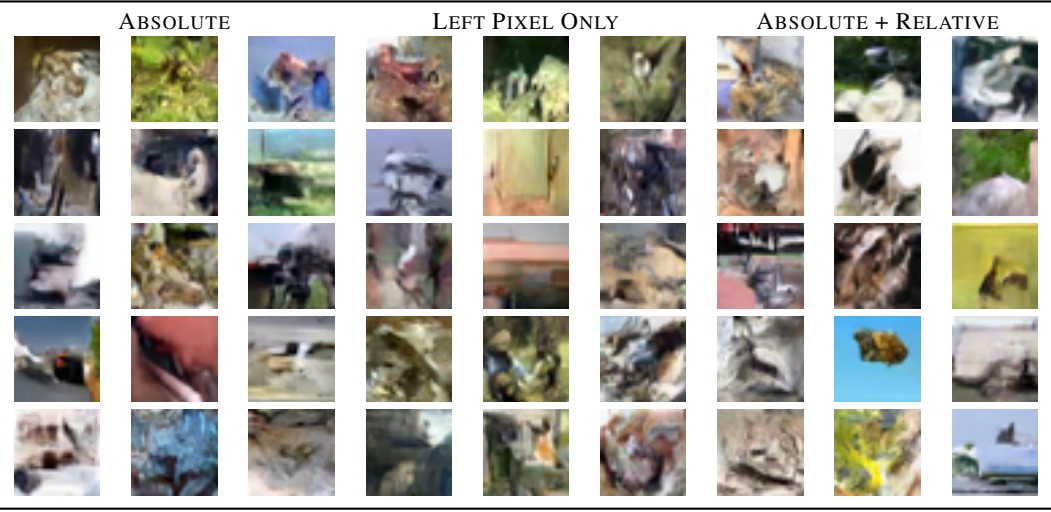

