# OpenReview forum: "Relative Pixel Prediction For Autoregressive Image Generation"
_ICLR.cc/2020/Conference — Reject_

### Official Review · AnonReviewer1 · 2019-10-22
**Official Blind Review #1**

**Rating:** 3

**Review:**

The paper proposes an approach for image generation that relies on an autoregressive model for the image pixels. These models are popularly used in image coding and compression settings, and have been used in generative models like PixelCNN. In contrast to this prior work, the proposed model is based on the selection of a previously available pixel and the modeling of the differences between the old pixel and the new one. The copy and adjustment models, i.e., eqs (3) and (5-6), are straightforward. Applications to image-to-image translation are also presented.

I am rating the paper "weak reject" mostly due to the limited set of comparisons in experimental results. There is no qualitative comparison to other algorithms for two of the problems considered (colorization, super-resolution) and the comparison with other algorithms for unconditional image generation is limited to CIFAR-10; thus, the impact of this contribution is not clear. Furthermore there is no discussion of these comparison results - i.e., what the proposed algorithm contributes given that it's outperformed by the sparse transformer.

It is not clear at first what the authors mean by "sub-pixel", which appears to be one of the color/spectrum channels of a pixel of the image? Also not clear what "outcome masking" refers to. The explanation of the hidden states (g,h) used for each mechanism are not always clear or explicit. For example, can you write an equation for h_{i,c-1} which is more explicit than "composing the history of generated sub-pixels"? Can you define Ui when it is first used in (6)? What is the difference between the pixel state h_{r,C} and its values x_r?

Minor comments
The second equation in Section 2.3 is missing =
In Section 5.1, it is not clear what is meant by "discrediting" the image.
The table in Fig. 3 could use full names for the problems instead of initials.

**Experience Assessment:**

I have published one or two papers in this area.

**Review Assessment: Checking Correctness Of Derivations And Theory:**

I assessed the sensibility of the derivations and theory.

**Review Assessment: Checking Correctness Of Experiments:**

I carefully checked the experiments.

**Review Assessment: Thoroughness In Paper Reading:**

I read the paper thoroughly.

---

### Official Review · AnonReviewer3 · 2019-10-22
**Official Blind Review #3**

**Rating:** 3

**Review:**

In this paper the authors present a new way to use autoregressive modeling to generate images pixel by pixel where each pixel is generated by modeling the difference between the current pixel value and  the preexistent ones. In order to achieve that, the authors propose a copy and adjustment mechanism that select an existing pixel, and then adjust its sub-pixel (channel values) to generate the new pixel. The proposed model is demonstrated with a suite of experiments in classic image generation benchmark. The authors also demonstrate the use of their technique in Image to Image translation.
Overall, although the paper explain clearly the intuition and the motivation of the proposed technique, I think that the paper in its present state have low novelty, weak related work analysis review and insufficient experiments to support a publication at ICLR.



**Novelty, contribution and related work**
The authors should highlight better their main contribution novelty of the proposed method compared to their baseline.


**Result and conducted experiments**
the correctness of the proposed approach is not proved by the conducted experiment  in fact:
The experiments do not provide the details of the used architecture compared to your baseline.
In Table 1 you report the results using your technique on several computer vision tasks (generation, colorization and super-resolution) but you're not comparing with the SoA of each of these tasks.
The  results reported in Tables 1 and 2 are not convincing  when compared to existing approaches (using only CIFAR10 in Table2).
There are so many missing details specially to validate Image-To-image translation
Figure 3 is confusing and  not clear

**Minor comments**
In  references section : (Kingma & Dhariwal, 2018) is not in a proper format (nips 2018)
Bad quality of illustrations and images
Be coherent with the position of captions (figure 3)

**Experience Assessment:**

I have read many papers in this area.

**Review Assessment: Checking Correctness Of Derivations And Theory:**

I carefully checked the derivations and theory.

**Review Assessment: Checking Correctness Of Experiments:**

I assessed the sensibility of the experiments.

**Review Assessment: Thoroughness In Paper Reading:**

I read the paper at least twice and used my best judgement in assessing the paper.

---

### Official Review · AnonReviewer4 · 2019-11-06
**Official Blind Review #4**

**Rating:** 3

**Review:**

The paper bases its methodology on well known developments in image analysis/synthesis about similarity of pixel values in adjacent locations. Many techniques have been used for modelling this similarity, including predictive models, cliques and graphs. The paper uses a simple autoregressive model for generating pixel values based on the values of previously processed pixels, estimating the differences between these neighboring pixel values.

The method is implemented through copying the pixel values and adjusting the differences.  Three types of prediction, based on absolute, or relative values are examined, for image generation, colorization, super-resolution. The problems are significant, but the approach rather superficial. A small experimental study is presented, based on CIFAR-10 and downsampled ImageNet datsaets. Much more experiments, including quantitative and qualitative results are reuired, to validate the prospects of the method in different types of (complex) problems and contexts. Marginal improvements are observed in the presented results. Since image generation and image to image translation are targeted, comparison and/or combined use with Sota methods, i.e., GANs should be examined.

Moreover, the paper presentation needs improvement; for example, symbols are undefined when used for the first time in the text (see eq. 3), etc.



**Experience Assessment:**

I have published in this field for several years.

**Review Assessment: Checking Correctness Of Derivations And Theory:**

I carefully checked the derivations and theory.

**Review Assessment: Checking Correctness Of Experiments:**

I carefully checked the experiments.

**Review Assessment: Thoroughness In Paper Reading:**

I read the paper thoroughly.

---

### Decision · Program_Chairs · 2019-12-19

**Decision:**

Reject

**Comment:**

All reviewers rated this submission as a weak reject and there was no author response.
The AC recommends rejection.